# Research on Identifying Robot Collision Points in Human–Robot Collaboration Based on Force Method Principle Solving

Zhijun Wang [1,2], Bocheng Zhu [1,2,*], Yue Yang [1,2] and Zhanxian Li [1,2]

1 College of Mechanical Engineering, North China University of Science and Technology, Tangshan 063210, China
2 Hebei Province Research Institute of Industrial Robot Industry Technology, Tangshan 063210, China
* Correspondence: zhubc@stu.ncst.edu.cn; Tel.: +86-139-3360-6536

**Abstract:** After years of more rigid and conventional production processes, the traditional manufacturing industry has been moving toward flexible manufacturing and intelligent manufacturing in recent years. After more than half a century of development, robotics has penetrated into all aspects of human production and life, bringing significant changes to the development of human society. At the same time, the key technology of human–machine cooperative operation has become a research hotspot, and how to realize a human–machine integrated safety system has attracted attention. Human–machine integration means that humans and robots can work in the same natural space, coordinating closely and interacting naturally. To realize real human–robot integration under human–robot cooperative operation, the good judgment of intentional interaction and accidental collision and the detection of collision joints and collision points when accidental collision occurs are the key points. In this paper, we propose a method to identify the collision joints by detecting real-time current changes in each joint of the robot and solve the collision point location information of the collision joints through the principle of virtual displacement and the principle of force method using the force sensor data installed at the base of the robot as the known condition. The results show that the proposed method of identifying the collision joints using changes in joint current and then establishing a collision detection algorithm to solve the collision point location is correct and reliable.

**Keywords:** human–machine collaboration; force method solution; collision point identification



## 1. Introduction

With the development of basic theories and key technologies for human–machine collaboration and human–machine integration, research focusing on the scientific issues of "human–machine–environment multimodal perception and interaction" is at a critical stage. The construction of a human–machine and machine–machine interaction system can promote the progress of robotics and theory while also meeting the needs of contemporary intelligent development [1–3]. Establishing collision detection models, deriving force control algorithms during robot motion, and establishing robot collision recognition and fast response mechanisms are the keys to ensuring safety during human–machine collaboration [4–8].

Common collision detection methods include installing optoelectronic, force-controlled, and joint torque sensors in specific parts of the robot [9–12] or detecting the relative position relationship between robot and human through vision systems, imaging systems, ranging systems, and envelope box methods [13–15] to ensure the safety of human–robot collaboration systems through pre-collision safety mechanisms. However, some of the systems are more complex in composition, have various types of sensors, have poor backward compatibility, have long fusion response time, and incur expensive system costs, thus making them not suitable for large-scale dissemination and use. Bharadwaja et al. [16] proposed a neural

network-based path planning control system for service robots, which was compared with the traditional probabilistic roadmap (PRM) algorithm, and the robot's obstacle distance prediction accuracy improved by about 36%. Meanwhile, the vision system can be used to model the surrounding environment in 3D and a genetic algorithm can be used to generate the optimal choice for the robot's forward route. A. Lomakin et al. [17] proposed a rigid robot collision detection method considering collision and external force, which provides an evaluable expression that can estimate the calculation error caused by numerical error and parameter uncertainty, which was applied in research into human–robot collaboration and robot safety. T. Lindner et al. [18] proposed a Reinforcement Learning (RL)-based collision avoidance strategy, which controls the robot's movement through trial-and-error interaction with the environment and achieves the goal of obstacle avoidance through the combination of the Deep Deterministic Policy Gradient (DDPG) algorithm and Hindsight Experience Replay (HER) algorithm. Alchan Yun et al. [19] designed a strain gauge-based cylindrical three-axis sensor to detect external force information by connecting the sensor to the robot, and it was experimentally verified that measuring force information with this structure of sensor can improve the efficiency of collision detection. In the literature [20,21], by installing color spheres on human shoulders, the head, limbs, etc., the camera IP captures data information to estimate human pose and position in performing the HRC task and uses the DLT method to estimate the 3D information of color spheres on the human body to avoid collisions through the use of image processing techniques. S. A. Gadsden et al. [22] proposed the smooth variable structure filter (SVSF), which has stronger accuracy and robustness in collision recognition and fault diagnosis for human–computer interaction systems with uncertainty in modeling and has been applied in the field of aerospace brakes.

Compared with the use of vision sensors, multi-sensor fusion, and energy drive for collision detection, the use of force sensors for collision recognition research has the advantages of direct measurement, simple algorithms, conduciveness to decoupling, and real-time response and is an important detection method in the field of human–machine integration. In this paper, we propose a collision detection algorithm based on the numerical information collected by a six-axis force/torque sensor installed at the base of a robot after static/dynamic force compensation and apply the force law principle and the virtual displacement principle to lock the location information of the collision point when the collision joint information is identified by real-time feedback of the current situation of each joint of the robot. Finally, the force compensation algorithm and collision detection algorithm are experimentally verified by carrying the experimental platform.

## 2. Model Building

In order to complete the collision detection task, a six-axis force/torque sensor was mounted at the base of the robot, namely the KWR200X. It is a large-range multi-dimensional force sensor with a built-in high-precision embedded data acquisition system, which can measure and transmit the orthogonal force $\boldsymbol{F}(F_x, F_y, F_z)$ and orthogonal moment $\boldsymbol{M}(M_x, M_y, M_z)$ in three directions in real time. It is made of high-strength alloy steel with strong bending resistance. The assembly method is bolted to the table and the robot is bolted to the six-dimensional force cell for better monitoring of experimental data. We define the robot base coordinate system as $O_0 - X_0Y_0Z_0$, where the $Z_0$ direction is opposite to the gravity direction and vertical upward. We define the six-axis force/torque sensor coordinate system as $O_s - X_sY_sZ_s$, and the six-axis force/torque sensor coordinate system can be obtained by rotating the robot base coordinate system around the $X_0$ axis by the $\theta$ angle; the attitude transformation matrix is shown in Equation (1), and the position relationship is shown in Figure 1.

$$ {}_{s}^{0}R = \begin{bmatrix} 1 & 0 & 0 \\ 0 & cos\theta & -sin\theta \\ 0 & sin\theta & cos\theta \end{bmatrix} \tag{1} $$

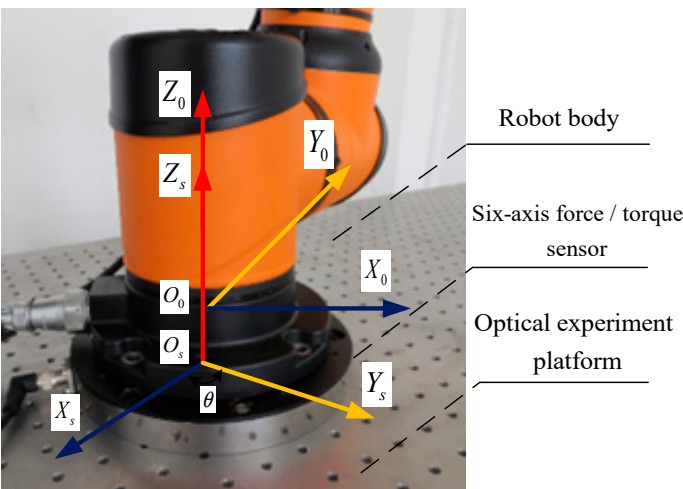

**Figure 1.** Relative position of the six-axis force/torque sensor's mounting position and coordinate system to the base coordinate system.

When the robot is working normally, each linkage is a common static structure, and the forces are fixed at both ends. When the robot linkage is collided with by the external environment, if the detected external force is defined as $F_i$, the original stationary structure becomes three-time super-stationary. The six-axis force/torque sensor installed at the robot base can measure the orthogonal forces and orthogonal moments in three directions: $F(F_x, F_y, F_z)$ and $M(M_x, M_y, M_z)$, respectively.

Therefore, the cosine of the magnitude and direction of the combined force is

$$F_i = \sqrt{F_x^2 + F_y^2 + F_z^2} \tag{2}$$

$$\begin{cases} \alpha = cos(F_i, i) = \frac{F_x}{F_i} \\ \beta = cos(F_i, j) = \frac{F_y}{F_i} \\ \gamma = cos(F_i, k) = \frac{F_z}{F_i} \end{cases} \tag{3}$$

The location information of the collision point can be expressed as

$$M = PF \tag{4}$$

$P(P_x, P_y, P_z)$ in Equation (4) is the position vector of the collision point relative to the six-axis force/torque sensor coordinate system. According to the vector information about force and moment in Equation (4), it is not possible to lock the collision point position coordinates, and other methods are needed to complete the task of collision point identification.

As shown in Figure 1, the directions of the three axes in the six-axis force/torque sensor coordinate system are known, and when a collision occurs, the robot linkage coordinate system needs to be re-established based on the original six-axis force/torque sensor coordinate system. We define the robot linkage coordinate system as $O_n - X_n Y_n Z_n$, and the robot linkage coordinate system can be rotated by the six-axis force/torque sensor coordinate system around $X_s$ by the $\alpha'$ angle first, then around the $Y_s$ axis by the $\beta'$ angle, and finally around $Z_s$ by the $\gamma'$ angle, and its attitude transformation matrix is shown in Equation (5).

$${}^s_n R_{X_s Y_s Z_s}(\alpha', \beta', \gamma') = R_{Z_s}(\gamma') R_{Y_s}(\beta') R_{X_s}(\alpha') = \begin{bmatrix} c\alpha'c\beta' & c\alpha's\beta' - s\alpha'c\gamma' & c\alpha's\beta'c\gamma' + s\alpha's\gamma' \\ s\alpha'c\beta' & s\alpha's\beta's\gamma' + c\alpha'c\gamma' & s\alpha's\beta'c\gamma' - c\alpha's\gamma' \\ -s\beta' & c\beta's\gamma' & c\beta'c\gamma' \end{bmatrix} \tag{5}$$

Here, $s\alpha$ stands for the abbreviation of $sin\alpha$ and $c\alpha$ stands for the abbreviation of $cos\alpha$. The presence of the tilt angle of the collaborative robot base mounting and the zero value of the six-axis force/torque sensor will provide the difference between the force information measured by the six-axis force/torque sensor and the actual magnitude of the external force of the collision. In the literature [23], the above compensation method is verified, where the collaborative robot base mounting inclination angle $\theta$ can be found. The coordinates at each joint are known at the factory when the robot is shipped, and when combined with the positions of each coordinate system defined above, the angles $\alpha'$, $\beta'$, and $\gamma'$ can be found. According to the Cartesian coordinate transformation rule, the magnitude of each directional component force measured by the six-axis force/torque sensor can be expressed in the linkage coordinate system as

$$\begin{aligned}
{}^nF_x &= {}^s_nR_{X_sY_sZ_s}(\alpha',\beta',\gamma')^T F_x \\
{}^nF_y &= {}^s_nR_{X_sY_sZ_s}(\alpha',\beta',\gamma')^T F_y \\
{}^nF_z &= {}^s_nR_{X_sY_sZ_s}(\alpha',\beta',\gamma')^T F_z
\end{aligned} \tag{6}$$

The $X_n$ direction is perpendicular to axis $i$ and the plane in which the connecting rod $i$ is located, the $Y_n$ direction is parallel to the connecting rod $i$ and points to axis $i+1$, and the $Z_n$ direction is parallel to axis $i$ and the plane in which the connecting rod $i$ is located.

As shown in Figure 2, the robot connecting rod $i$ is subjected to the external force $F_i$ after the collision of the robot connecting rod $i$. The effect of the force in the direction of ${}^nF_x$ is the same as the effect of the force in the direction of ${}^nF_z$. When considering the force on rod $i$, the effect on ${}^nF_z$ is used as an example to derive the result, and the algebraic summation of the two parts is performed.

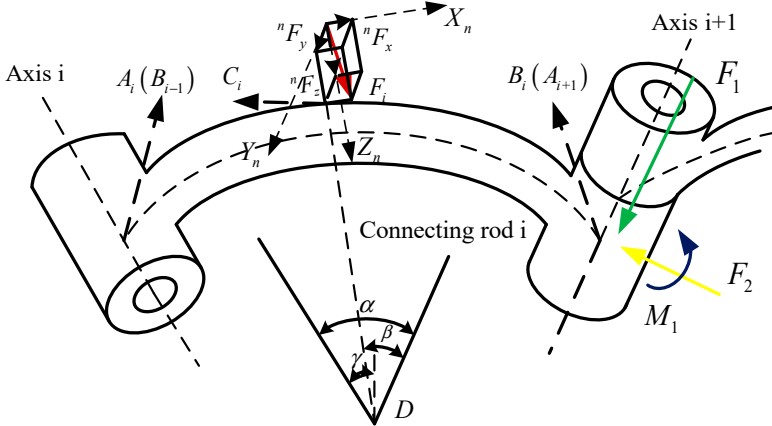

**Figure 2.** Collision force schematic of robot linkage $i$ with external force $F_i$.

In the figure, point $C_i$ is defined as the position of the collision point where force $F_i$ collides with robot rod $i$; point $D$ is defined as the intersection point of axis $i$ and the axis $i+1$ extension line; $a$ is the distance between axis $i$ and axis $i+1$ center-of-mass positions $A_i$ and $B_i$ to point $D$; $\alpha$ is the angle between the two extension lines; $\beta$ is the angle between a line in the plane where forces ${}^nF_z$ and ${}^nF_x$ are located and the axis $i+1$ angle of the extension line; and $\gamma$ is an angle variable in the force method to solve the super-stationary structure.

## 3. Force Compensation Algorithm

When the robot is at rest, the six-axis force/torque sensor mounted at the base will output different values in each direction due to the gravity of the body in different positions. Therefore, in order to identify the collision point at rest, the gravity should be compensated to achieve a zero reading of the six-axis force/torque sensor mounted at the base when the robot is at rest. Therefore, when the sensor value changes irregularly, it is assumed that an accidental collision has occurred between the robot and the external environment.

To establish the position and transformation relationship between the adjacent links of the collaborative robot using the "D − H" parameter method, the parameters and variables of each link should be defined first. $\alpha_{i-1}$ is the angle of rotation from $Z_{i-1}$ to $Z_i$ around the $X_{i-1}$ axis; $a_{i-1}$ is the distance from $Z_{i-1}$ to $Z_i$ along the $X_{i-1}$ axis; $d_i$ is the distance from $X_{i-1}$ to $X_i$ along the $Z_i$ axis; and $\theta_i$ is the angle of rotation from $X_{i-1}$ to $X_i$ around the $Z_i$ axis. Figure 3 shows the "D − H" coordinate space of the two adjacent joints.

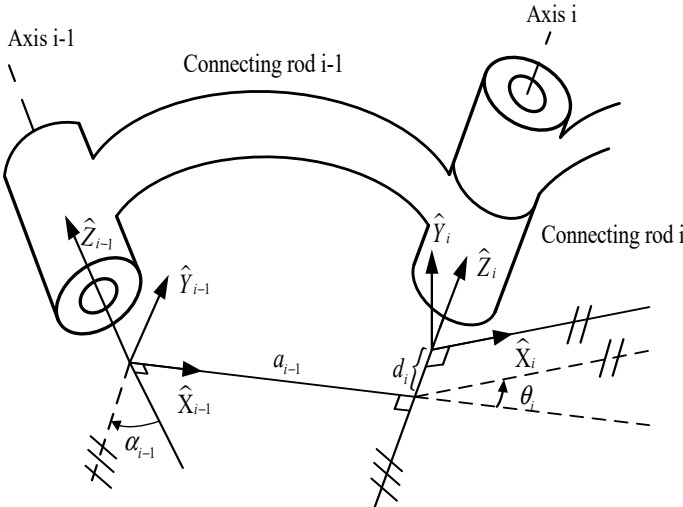

**Figure 3.** "D − H" coordinate space of two adjacent joint links.

In the robot linkage coordinate system, the coordinate system $\{i-1\}$ can be obtained by four flush transformations of the coordinate system $\{i\}$; therefore, the flush transformation matrix between two adjacent linkage coordinate systems $\{i-1\}$ and $\{i\}$ is

$$
^{i-1}T_i = Rot(X, \alpha_{i-1})Trans(X, a_{i-1})Trans(Z, d_i)Rot(Z, \theta_i) =
$$

$$
\begin{bmatrix} 1 & 0 & 0 & 0 \\ 0 & c\alpha_{i-1} & -s\alpha_{i-1} & 0 \\ 0 & s\alpha_{i-1} & c\alpha_{i-1} & 0 \\ 0 & 0 & 0 & 1 \end{bmatrix}
\begin{bmatrix} 1 & 0 & 0 & a_{i-1} \\ 0 & 1 & 0 & 0 \\ 0 & 0 & 1 & 0 \\ 0 & 0 & 0 & 1 \end{bmatrix}
\begin{bmatrix} 1 & 0 & 0 & 0 \\ 0 & 1 & 0 & 0 \\ 0 & 0 & 1 & d_i \\ 0 & 0 & 0 & 1 \end{bmatrix}
\begin{bmatrix} c\theta_i & -s\theta_i & 0 & 0 \\ s\theta_i & c\theta_i & 0 & 0 \\ 0 & 0 & 1 & 0 \\ 0 & 0 & 0 & 1 \end{bmatrix} =
$$

$$
\begin{bmatrix}
c\theta_i & -s\theta_i & 0 & a_{i-1} \\
s\theta_i c\alpha_{i-1} & c\theta_i c\alpha_{i-1} & -s\alpha_{i-1} & -d_i s\alpha_{i-1} \\
s\theta_i s\alpha_{i-1} & c\theta_i s\alpha_{i-1} & c\alpha_{i-1} & d_i c\alpha_{i-1} \\
0 & 0 & 0 & 1
\end{bmatrix}
\tag{7}
$$

By bringing the parameters of each linkage of the robot into Equation (7), the positional transformation matrix of the specified linkage can be obtained. Here, the transformation matrix of the coordinate system $\{i\}$ with respect to the base coordinate system can be expressed as

$$
^0_i T = {}^0_1 T {}^1_2 T {}^2_3 T \cdots {}^{i-2}_{i-1} T {}^{i-1}_i T =
\begin{bmatrix}
n_x & o_x & a_x & p_x \\
n_y & o_y & a_y & p_y \\
n_z & o_z & a_z & p_z \\
0 & 0 & 0 & 1
\end{bmatrix}
\tag{8}
$$

where ${}^{i-1}_i R = \begin{bmatrix} n_x & o_x & a_x \\ n_y & o_y & a_y \\ n_z & o_z & a_z \end{bmatrix}$ denotes the rotation matrix of the coordinate system $\{i\}$ with respect to the coordinate system $\{i-1\}$, and ${}^{i-1}_i p = \begin{bmatrix} p_x & p_y & p_z \end{bmatrix}^T$ denotes the position matrix of the coordinate system $\{i\}$ with respect to the coordinate system $\{i-1\}$.

We define the position vector of the center of mass $c_i$ of any linkage $i$ with respect to the joint coordinate system $\{i\}$ as

$$c_i = \begin{bmatrix} c_{ix} & c_{iy} & c_{iz} \end{bmatrix}^T \qquad (9)$$

The transformation matrix of the connecting rod coordinate system $\{i\}$ with respect to the base coordinate system can be represented by ${}^0_i R$. The connecting rod center-of-mass coordinate system $\{c_i\}$ has the same direction as the coordinate system $\{i\}$, and its position relationship is shown in Figure 1. If the gravity vector of a robot linkage on its center-of-mass coordinate system is

$$G_i = \begin{bmatrix} 0 & 0 & G_i \end{bmatrix}^T \qquad (10)$$

then the chi-square transformation matrix of the linkage's center-of-mass coordinate system $\{c_i\}$ with respect to the base coordinate system is

$$ {}^0_{c_i} T = {}^0_i T\, {}^i_{c_i} T = \begin{bmatrix} {}^0_{c_i} R & {}^0_{c_i} p \\ 0 & 1 \end{bmatrix} \qquad (11) $$

where ${}^0_{c_i} p = \begin{bmatrix} p_{cx} & p_{cy} & p_{cz} \end{bmatrix}^T$.

Then, when the robot is in the static state, its own gravity vector, when transformed from the linkage coordinate system to the base coordinate system, is expressed by ${}^0 f$ as

$$ {}^0 f = \begin{bmatrix} 0 & 0 & \displaystyle\sum_{i=0}^{n} G_i \end{bmatrix}^T \qquad (12) $$

Its moment vector, when converted from the linkage coordinate system to the base coordinate system, is expressed in ${}^0 m$ as

$$ {}^0 m = \begin{bmatrix} -\displaystyle\sum_{i-1}^{n}(G_i \times p_{cy}) & -\displaystyle\sum_{i-1}^{n}(G_i \times p_{cx}) & 0 \end{bmatrix}^T \qquad (13) $$

Thus, the transformation matrices of the force and moment vectors in the force sensor coordinate system and the base coordinate system can be obtained as

$$ \begin{bmatrix} {}^s f \\ {}^s m \end{bmatrix} = \begin{bmatrix} {}^s_o R & 0 \\ {}^s P_{0ORG}\, {}^s_o R & {}^s_o R \end{bmatrix} \begin{bmatrix} {}^o f \\ {}^o m \end{bmatrix} \qquad (14) $$

where ${}^s_o R = \begin{bmatrix} 1 & 0 & 0 \\ 0 & 1 & 0 \\ 0 & 0 & 1 \end{bmatrix}$; ${}^s P_{0ORG} = \begin{bmatrix} 0 & -p_z & p_y \\ p_z & 0 & -p_x \\ -p_y & p_x & 0 \end{bmatrix} = \begin{bmatrix} 0 & -h & 0 \\ h & 0 & 0 \\ 0 & 0 & 0 \end{bmatrix}$.

The robot's speed and acceleration change over time during normal operation, so gravity compensation alone cannot meet the requirements. After the dynamic force compensation, the sensor reading will be zero, so that if a collision occurs, the collision point can be identified and calculated using the compensated data.

The Newton–Euler method can build a rigid robot dynamics model using only the velocity, angular velocity, and rotational inertia of each linkage of the robot. The velocity and acceleration of each linkage of the robot are obtained by first pushing outward from the robot base and then pushing inward from the outside to obtain the generalized force of interaction between two adjacent linkages.

For a single operator arm linkage $i$ of the robot, the forces are shown in Figure 4.

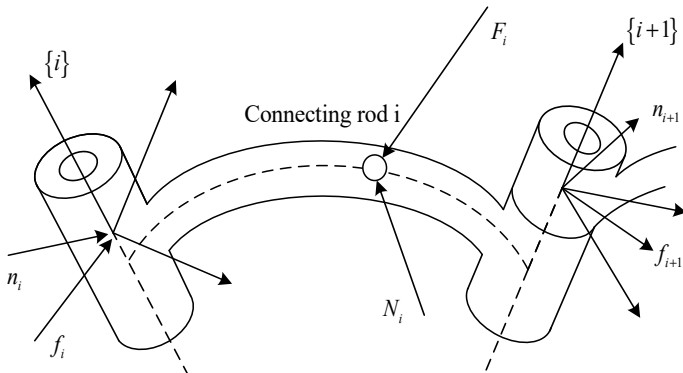

**Figure 4.** Robot linkage force diagram.

We define the mass of each operating arm of the robot as $m_i (i = 1, 2, 3, \cdots, 6)$, and the position vector of the center of mass $c_i$ of robot joint $i$ with respect to the joint coordinate system $\{i\}$ can be expressed as $r_i (i = 1, 2, 3, \cdots, 6)$. We define the displacement of robot joint $i$ as $\theta_i$, the velocity as $\dot{\theta}_i$, and the acceleration as $\ddot{\theta}_i$. The extrapolation process according to Equation (15) is mainly to solve for the acceleration of the center of mass of each joint of its robot.

$$
\begin{cases}
{}^{i+1}\omega_{i+1} = {}^{i+1}_{i}R\,{}^{i}\omega_i + \dot{\theta}_{i+1}{}^{i+1}Z_{i+1} \\
{}^{i+1}\dot{\omega}_{i+1} = {}^{i+1}_{i}R\,{}^{i}\dot{\omega}_i + {}^{i+1}_{i}R\,{}^{i}\omega_i \times \dot{\theta}_{i+1}{}^{i+1}Z_{i+1} + \ddot{\theta}_{i+1}{}^{i+1}Z_{i+1} \\
{}^{i+1}\dot{v}_{i+1} = {}^{i+1}_{i}R\left[{}^{i}\dot{\omega}_i \times {}^{i}P_{i+1} + {}^{i}\omega_i \times \left({}^{i}\omega_i \times {}^{i}P_{i+1}\right) + {}^{i}\dot{v}_i\right] \\
{}^{i}\dot{v}_{c_i} = {}^{i}\dot{\omega}_i \times {}^{i}r_{i+1} + {}^{i}\omega_i \times \left({}^{i}\omega_i \times {}^{i}r_{i+1}\right) + {}^{i}\dot{v}_i
\end{cases}
\tag{15}
$$

where $\omega_i$, $\dot{\omega}_i$, $\dot{v}_i$ and $\dot{v}_{c_i}$ are the angular velocity, angular acceleration, linear acceleration, and center-of-mass linear acceleration of the robot linkage $i$, respectively; $\omega_0 = \dot{\omega}_0 = v_0 = \dot{v}_0 = \begin{bmatrix} 0 & 0 & 0 \end{bmatrix}^{\mathrm{T}}$; $Z_i (i = 1, 2, \cdots, 6) = \begin{bmatrix} 0 & 0 & 0 \end{bmatrix}^{\mathrm{T}}$.

Accordingly, Newton's equation and Euler's equation of motion can be established:

$$
{}^{i-1}F_i = {}^{i-1}F_i + m_i\ddot{v}_{c_i} - {}^{c_i}G_i
\tag{16}
$$

$$
{}^{i-1}M_i = {}^{i-1}M_i - r_{i+1,c_i} \times {}^{i+1}F_i + r_{i,c_i} \times {}^{i+1}F_i + I_i\dot{\omega}_i + \omega_i \times I_i\omega_i
\tag{17}
$$

where ${}^{i-1}F_i$ and ${}^{i-1}M_i$ denote the force and moment of the robot linkage $i-1$ on the linkage $i$, respectively; ${}^{i+1}F_i$ and ${}^{i+1}M_i$ denote the force and moment of the robot linkage $i+1$ on the linkage $i$, respectively; $r_{i,c_i}$ and $r_{i+1,c_i}$ denote the position vector from the coordinate origin to the center of mass $c_i$ attached on robot joints $i$ and $i+1$, respectively; $I_i$ denotes the inertia tensor of robot linkage $i$ with respect to the center of mass $c_i$; and $\omega_i \times I_i\omega_i$ denotes the Coe-style force term.

For the n-degree-of-freedom robot, when there is no load at its end, the extrapolated recursive formulas from Equations (15)–(17) can calculate $F_o^1 = -F_1^o$ and $M_o^1 = -M_1^o$, which are then carried over to gravity compensation Formula (12) to (14) in the stationary state. The required gravity compensation values for the six-axis force/torque sensor at the base, namely ${}^S f$ and ${}^S M$, can be solved for when the robot is in motion.

When the robot is in normal operation, the six-dimensional force sensor installed at the base reads the force and moment information as ${}^D f$ and ${}^D M$. The force and moment relationship information obtained after gravity compensation is $F$ and $M$, followed by

$$
{}^{i-1}M_i = {}^{i-1}M_i - r_{i+1,c_i} \times {}^{i+1}F_i + r_{i,c_i} \times {}^{i+1}F_i + I_i\dot{\omega}_i + \omega_i \times I_i\omega_i
\tag{18}
$$

Thus, when the robot is in normal operation, the six-axis force/torque sensor mounted at the base of the robot reads zero after dynamic force compensation. Then, when the value of the six-axis force/torque sensor exceeds a set threshold, a collision between the

robot and the external environment is considered to have occurred. Based on the six-axis force/torque sensor and the working principle of the robot itself, the spatial coordinates at the front axis of the collision linkage and the spatial coordinates at the rear axis of the collision linkage at the time of collision are

$$\begin{cases} P_1(x_1, y_1, z_1) \\ P_2(x_2, y_2, z_2) \end{cases} \tag{19}$$

## 4. The Principle of Force Method to Solve the Location of the Collision Point

When the joints where the robot collides are known, the forces at robot joint axes $i$ and $i + 1$ due to the collision are shown in Figure 5. We define the point $C_i$ as the location of the collision point where the force $F_i$ collides with the robot joint $i$, we define $l$ as the length of the robot joint $i$, and we define $l'$ as the length of the robot axis $i + 1$ to the cross section where the collision point is located and the variables in the force law principle.

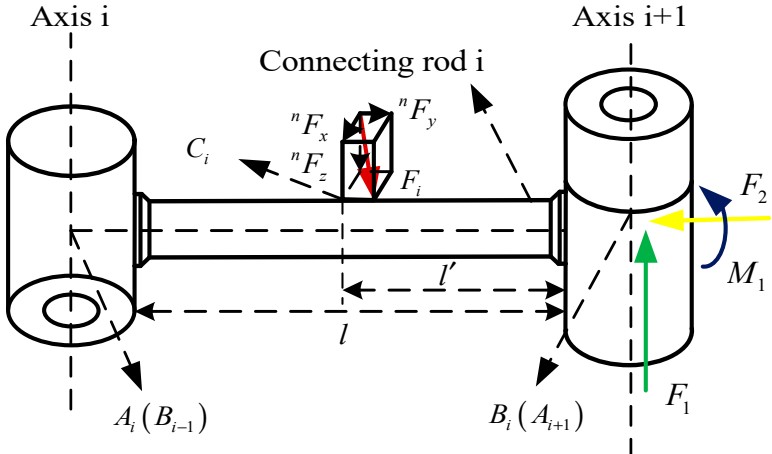

**Figure 5.** Collision force schematic of robot linkage $i$ subjected to external force $F_i$.

When applying the principle of force method to solve the position of the collision point, the $B_i$ end of the connecting rod $i$ where the collision occurs is first released, and the original super-stationary structure becomes a basic static system. In this static system, in addition to the action of $^nF_z$, the vertical force $F_1^z$, the horizontal force $F_2^z$, and the moment $M_1^z$, which are in the same direction as axis $i - 1$ and perpendicular to axis $i$, act at the end of $A_{i+1}$, as shown in Figure 6.

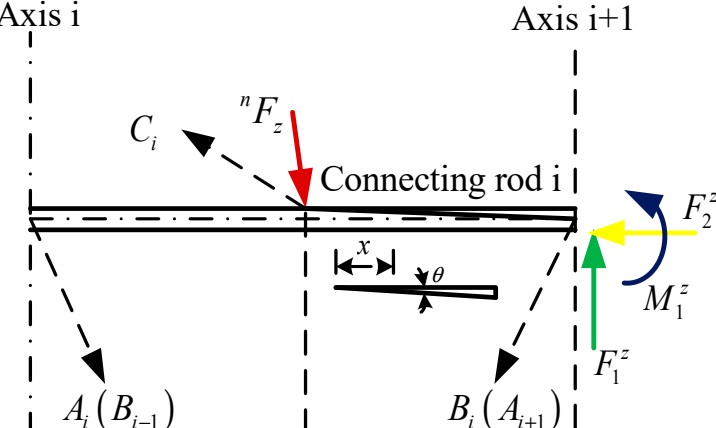

**Figure 6.** Schematic diagram of robot linkage $i$ after unconstraint under the action of $^nF_z$.

The robot linkage is a linear elastic structure, so the displacement is proportional to the force. $\Delta_{1F}$ denotes the displacement of the $B_i$ end along the direction of $F_1^z$ under the force $^nF_z$, and $\delta_{11}, \delta_{12}$, and $\delta_{13}$ denote the displacement of the $B_i$ end of the robot linkage $i$ along the direction of $F_1^z$ when $F_1^z, F_2^z$, and $M_1^z$ are unit forces, respectively, with individual displacement acting on the robot linkage $i$. Thus, the total displacement of the $B_i$ end of the robot linkage $i$ along the $F_1^z$ direction is

$$\Delta_1 = \delta_{11}F_1^z + \delta_{12}F_2^z + \delta_{13}M_1^z + \Delta_{1F} \tag{20}$$

Since the $B_i$ end of the robot linkage $i$ is a fixed constraint, the displacement at the $B_i$ point along the $F_1^z$ direction should be 0, and the deformation coordination equation thus becomes

$$\Delta_1 = \delta_{11}F_1^z + \delta_{12}F_2^z + \delta_{13}M_1^z + \Delta_{1F} = 0 \tag{21}$$

Following the same method, two other deformation coordination equations can be obtained for the displacement of the robot linkage $B_i$ point along the $F_2^z$ direction equal to zero and the angle of rotation in the $M_1^z$ direction equal to zero. Thus, the linear equation equation is as follows:

$$\begin{cases} \delta_{11}F_1^z + \delta_{12}F_2^z + \delta_{13}M_1^z + \Delta_{1F} = 0 \\ \delta_{21}F_1^z + \delta_{22}F_2^z + \delta_{23}M_1^z + \Delta_{2F} = 0 \\ \delta_{31}F_1^z + \delta_{32}F_2^z + \delta_{33}M_1^z + \Delta_{3F} = 0 \end{cases} \tag{22}$$

The nine coefficients $\delta_{ij}(i = 1, 2, 3; j = 1, 2, 3)$ and three constant terms $\Delta_{iF}(i = 1, 2, 3)$ in Equation (22) are represented as shown in Figure 7. In the figures, (a)–(d) show the displacement diagrams of the $B_i$ end of the robot linkage along the $F_1^z, F_2^z$, and $M_1^z$ directions when $F_1^z, F_2^z, M_1^z$, and $^nF_z$ are unit forces.

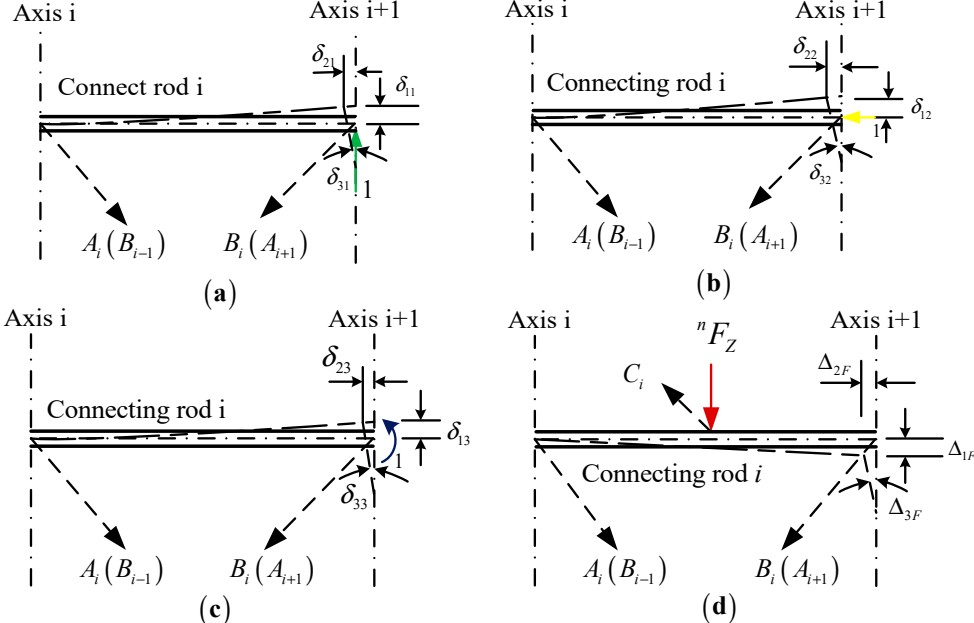

**Figure 7.** Diagram of the displacement along different directions when different forces are applied to the robot linkage. (**a**) Displacement diagram of the connecting rod $B_i$ along the $F_1^z, F_2^z$, and $M_1^z$ directions when the unit force is $F_1^z$. (**b**) Displacement diagram of the connecting rod $B_i$ along the $F_1^z$, $F_2^z$, and $M_1^z$ directions when the unit force is $F_2^z$. (**c**) Displacement diagram of the connecting rod $B_i$ along the $F_1^z, F_2^z$, and $M_1^z$ directions when the unit force is $M_1^z$. (**d**) $^nF$ plot of the displacement of the connecting rod $B_i$ along the $F_1^z, F_2^z$, and $M_1^z$ directions when the unit force is united.

When considering the impact of the collision process on the displacement of the robot linkage, the impact of the shear force and the axial force is much smaller than the impact of the moment and can be neglected. As shown in Figure 6, when the basic static system in

which the robot linkage $i$ is located only acts with the external force $^nF_z$, the moment on the robot linkage is

$$M = 0 \ (0 \leq x \leq l'), \ M = \frac{^nF_z(l - l')x}{l} \ (0 \leq x \leq l') \tag{23}$$

If the moment of the increase in the curvature of the robot linkage is specified as positive, the moment equation when a unit force is applied at point $B_i$ on the robot linkage along the direction of $F_1^z$ is

$$M_1 = -x \tag{24}$$

When a unit force is applied at point $B_i$ on the robot linkage along the $F_2^z$ direction, the moment equation is

$$M_2 = 0 \tag{25}$$

When a unit force is applied at point $B_i$ on the robot linkage along the direction $M_1^z$, the moment equation is

$$M_3 = -1 \tag{26}$$

According to the displacement reciprocity theorem, it is proved that $\delta_{ij} = \delta_{ji}(i, j = 1, 2, 3)$. Therefore, it is obtained that

$$\Delta_{1F} = \int_l \frac{MM_1}{EI} ds = \frac{-^nF_z(l - l')l'^3}{3EI} \tag{27}$$

$$\Delta_{2F} = \int_l \frac{MM_2}{EI} ds = 0 \tag{28}$$

$$\Delta_{3F} = \int_l \frac{MM_3}{EI} ds = \frac{-^nF_z(l - l')l'^2}{2EI} \tag{29}$$

$$\delta_{11} = \int_l \frac{M_1M_1}{EI} ds = \frac{l^4}{3EI} \tag{30}$$

$$\delta_{22} = \int_l \frac{M_2M_2}{EI} ds = 0 \tag{31}$$

$$\delta_{33} = \int_l \frac{M_3M_3}{EI} ds = \frac{l^2}{EI} \tag{32}$$

$$\delta_{12} = \delta_{21} = \int_l \frac{M_1M_2}{EI} ds = 0 \tag{33}$$

$$\delta_{13} = \delta_{31} = \int_l \frac{M_1M_3}{EI} ds = \frac{l^3}{2EI} \tag{34}$$

$$\delta_{23} = \delta_{32} = \int_l \frac{M_2M_3}{EI} ds = 0 \tag{35}$$

The system of Equation (22) is simplified to obtain the formula for the force-moment:

$$F_1^z = \frac{a_{11} + a_{12} + a_{13}}{b_{11}} \tag{36}$$

$$F_2^z = \frac{a_{21} + a_{22} + a_{23}}{b_{22}} \tag{37}$$

$$M_1^z = \frac{a_{31} + a_{32} + a_{33}}{b_{33}} \tag{38}$$

Among them,

$$a_{11} = \left(\delta_{23}{}^2 - \delta_{22}\delta_{23}\delta_{33}\right)\Delta_{1F},$$
$$a_{12} = \left(-\delta_{13}{}^2\delta_{12} + \delta_{12}\delta_{23}\delta_{33}\right)\Delta_{2F}, a_{13} = \left(-\delta_{23}{}^2\delta_{12} + \delta_{13}\delta_{22}\delta_{23}\right)\Delta_{3F}$$
$$b_{11} = 2\delta_{23}{}^2\delta_{12}\delta_{13} - \delta_{23}{}^3\delta_{11} - \delta_{12}{}^2\delta_{23}\delta_{33} + \delta_{11}\delta_{22}\delta_{33}\delta_{23} - \delta_{13}{}^2\delta_{22}\delta_{23}$$
$$a_{21} = \left(\delta_{12}{}^2\delta_{33} - \delta_{12}\delta_{13}\delta_{22}\right)\Delta_{1F}$$
$$a_{22} = \left(\delta_{13}{}^2\delta_{12} - \delta_{11}\delta_{12}\delta_{33}\right)\Delta_{2F}, a_{23} = \left(-\delta_{12}{}^2\delta_{13} + \delta_{11}\delta_{12}\delta_{23}\right)\Delta_{3F}$$
$$b_{22} = 2\delta_{12}{}^2\delta_{13}\delta_{23} - \delta_{23}{}^2\delta_{11}\delta_{12} - \delta_{12}{}^3\delta_{33} + \delta_{11}\delta_{22}\delta_{33}\delta_{12} - \delta_{13}{}^2\delta_{12}\delta_{22}$$
$$a_{31} = \left(\delta_{12}{}^2\delta_{23} - \delta_{12}\delta_{13}\delta_{22}\right)\Delta_{1F}$$
$$a_{32} = \left(\delta_{12}{}^2\delta_{13} - \delta_{11}\delta_{12}\delta_{23}\right)\Delta_{2F}, a_{33} = \left(-\delta_{12}{}^2 + \delta_{11}\delta_{12}\delta_{22}\right)\Delta_{3F}$$
$$b_{33} = \delta_{13}{}^2\delta_{12}\delta_{22} - 2\delta_{12}{}^2\delta_{13}\delta_{23} + \delta_{23}{}^2\delta_{11}\delta_{12} + \delta_{12}{}^3\delta_{33} - \delta_{11}\delta_{22}\delta_{33}\delta_{12}$$

When solving the effect of the robot linkage due to the action of ${}^nF_x$, the method is exactly the same as the calculation and solution method for solving ${}^nF_z$. When ${}^nF_z$ and ${}^nF_x$ act on the robot connecting rod at the same time when establishing the connecting rod coordinate system, the two sets of forces are in two different planes after being unconstrained. The force-moments generated under the action of ${}^nF$ are $F_1^z$, $F_2^z$, and $M_1^z$, and the force-moments generated under the action of ${}^nF_x$ are $F_1^x$, $F_2^x$, and $M_1^x$, with the two sets of force-moments algebraically summed as $F_1$, $F_2$, and $M_1$. Therefore, when a collision occurs, the effect of the external force on the robot joint axis $i$ is shown in Figure 5. In practical applications, when the software and hardware of the robot system are installed, the program is run. First, the collision joint is locked according to the change in the output joint current. Combined with the force and moment values output by the robot during normal operation and some parameter information for the robot body, it is brought into the Equations (23)–(35) to solve the algorithm parameters. In Equations (36)–(38), because the value information of force and moment at the time of collision is known, only the only variable $l'$ in the algorithm is the unknown condition, and $l'$ and the collision point location $P$ can be found. At this point, the theoretical derivation of the force compensation algorithm and the principle of the force law, i.e., how the location of the collision point is solved, is completed.

## 5. Experimental Verification and Analysis of Results

### 5.1. Identification of Collision Joints for Experimental Verification

In the process of human–robot cooperative operation, the robot system information can be read by observing changes in the current of each joint of the robot and making a judgment as to whether the robot has had an accidental collision. To read the current data of each joint of the robot in real time, we can write a script program to write the robot system by making the robot communicate with the host computer through a network protocol to read the data. The main program is mainly divided into four parts: serial communication between the robot and the host computer, the assignment of values to each joint of the robot, loading real-time data for transfer to the host computer, and transmission of the data collected in the RET function to the host computer for display, of which part of the main program is shown below.

```
Joint_AO_1=get modbus io status("Joint_AO_1")
Joint_AO_2=get modbus io status("Joint_AO_2")
Joint_AO_3=get modbus io status("Joint_AO_3")
Joint_AO_4=get modbus io status("Joint_AO_4")
Joint_AO_5=get modbus io status("Joint_AO_5")
Joint_AO_6=get modbus io status("Joint_AO_6")
```

```
if (Joint_AO_1 > 30,000) then
Joint_current_1=Joint_AO_1-65535
end
if (Joint_AO_1 < 30,000) then
Joint_current_1=Joint_AO_1
end
if (Joint_AO_2 > 30,000) then
Joint_current_2=Joint_AO_2-65535
end
if (Joint_AO_2 < 30,000) then
Joint_current_2=Joint_AO_2
end
if (Joint_AO_3 > 30,000) then
Joint_current_3=Joint_AO_3-65535
end
if (Joint_AO_3 < 30,000) then
Joint_current_3=Joint_AO_3
end
if (Joint_AO_4 > 30,000) then
Joint_current_4=Joint_AO_4-65535
end
if (Joint_AO_4 < 30,000) then
Joint_current_4=Joint_AO_4
end
if (Joint_AO_5 > 30,000) then
Joint_current_5=Joint_AO_5-65535
end
if (Joint_AO_5 < 30,000) then
Joint_current_5=Joint_AO_5
end
if (Joint_AO_6 > 30,000) then
Joint_current_6=Joint_AO_6-65535
end
if (Joint_AO_6 < 30,000) then
Joint_current_6=Joint_AO_6
end
```

To run the program, you first need to understand the current values of each joint when the robot works normally. According to the output real-time current value of each joint, it is possible to determine whether an accidental collision has occurred. When an accidental collision occurs, the algorithm parameters are solved according to the parameter information of the robot body and the output value of the force sensor. Then, based on the value of the force sensor at the time of collision, the magnitude of variable l in the algorithm is solved. At this point, the location of the collision point can be represented. The algorithm block diagram is shown in Figure 8.

As shown in Figure 9a, volunteers simulated the collision experiment with their hands and applied external forces to robot joints 1 to 6, respectively. At the same time, the computer collected the current at each joint of the robot in six action states and output it to the computer in the form of numerical values. This current information is output in groups every 500 ms. Figure 9b shows a line chart corresponding to the output of the process of the volunteer hand simulation collision experiment in Figure 9a. The horizontal axis of this line chart is represented as a set of current information collected every 500 ms. The vertical axis of the line chart represents the real-time current situation of each joint of the robot corresponding to each set of current information.

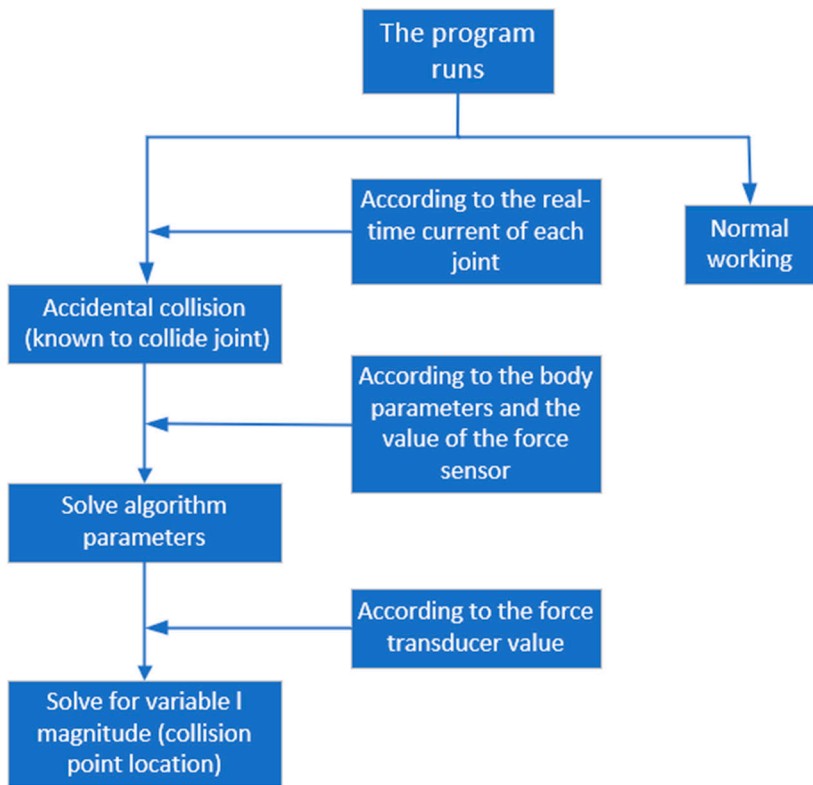

**Figure 8.** Diagram of the collision detection algorithm.

*5.2. Force Method Principle for Collision Point and Error Analysis*

The force method used the AUBO i5 series collaborative robot body parameters as input conditions and applied two external forces of "60N" and "100N" in various direction, as measured by a tensiometer tester. The calculated values are obtained by substituting the formula. The material of the robot linkage was the common carbon fiber "T300" and its elasticity model is $E$. The robot arm cross-section is circular and its moment of inertia is $I_z$. The numerical information of each parameter of the force principle is shown in Table 1.

**Table 1.** Information on the value of each parameter of the force law principle.

| Projects | Location Information/l′ | Split Force$^{n}F_z$/N | $\delta_{11}$ | $\delta_{22}$ | $\delta_{33}$ | $\delta_{12}=\delta_{21}$ | $\delta_{13}=\delta_{31}$ | $\delta_{23}=\delta_{32}$ | $\Delta_{1F}$ | $\Delta_{2F}$ | $\Delta_{3F}$ |
|---|---|---|---|---|---|---|---|---|---|---|---|
| 1 | 0.1 | 60 | 0.0031 | 0 | 0.1616 | 0 | 0.0194 | 0 | −0.0079 | 0 | −0.1178 |
| 2 | 0.1 | 100 | 0.0031 | 0 | 0.1616 | 0 | 0.0194 | 0 | −0.0131 | 0 | −0.1964 |
| 3 | 0.11 | 60 | 0.0031 | 0 | 0.1616 | 0 | 0.0194 | 0 | −0.0097 | 0 | −0.1324 |
| 4 | 0.11 | 100 | 0.0031 | 0 | 0.1616 | 0 | 0.0194 | 0 | −0.0162 | 0 | −0.2206 |
| 5 | 0.12 | 60 | 0.0031 | 0 | 0.1616 | 0 | 0.0194 | 0 | −0.0116 | 0 | −0.1454 |
| 6 | 0.12 | 100 | 0.0031 | 0 | 0.1616 | 0 | 0.0194 | 0 | −0.0194 | 0 | −0.2424 |
| 7 | 0.13 | 60 | 0.0031 | 0 | 0.1616 | 0 | 0.0194 | 0 | −0.0136 | 0 | −0.1564 |
| 8 | 0.13 | 100 | 0.0031 | 0 | 0.1616 | 0 | 0.0194 | 0 | −0.0226 | 0 | −0.2607 |
| 9 | 0.14 | 60 | 0.0031 | 0 | 0.1616 | 0 | 0.0194 | 0 | −0.0154 | 0 | −0.1649 |
| 10 | 0.14 | 100 | 0.0031 | 0 | 0.1616 | 0 | 0.0194 | 0 | −0.0256 | 0 | −0.2749 |

By bringing the values of the parameters in Table 1 into Equations (27)–(35), the numerical solution of the force generated by the robot linkage due to the collision action of $^{n}F_z$ and $^{n}F_x$ can be obtained. In order to verify the correctness and accuracy of the calculation procedure, an error analysis of the experimental results was performed.

The absolute error for the actual external contact force is calculated by the formula

$$\delta F = |\Delta F_s - \Delta F_c| \tag{39}$$

where $\Delta F_s$ is the experimental value of the external force and $\Delta F_c$ is the calculated value of the external force.

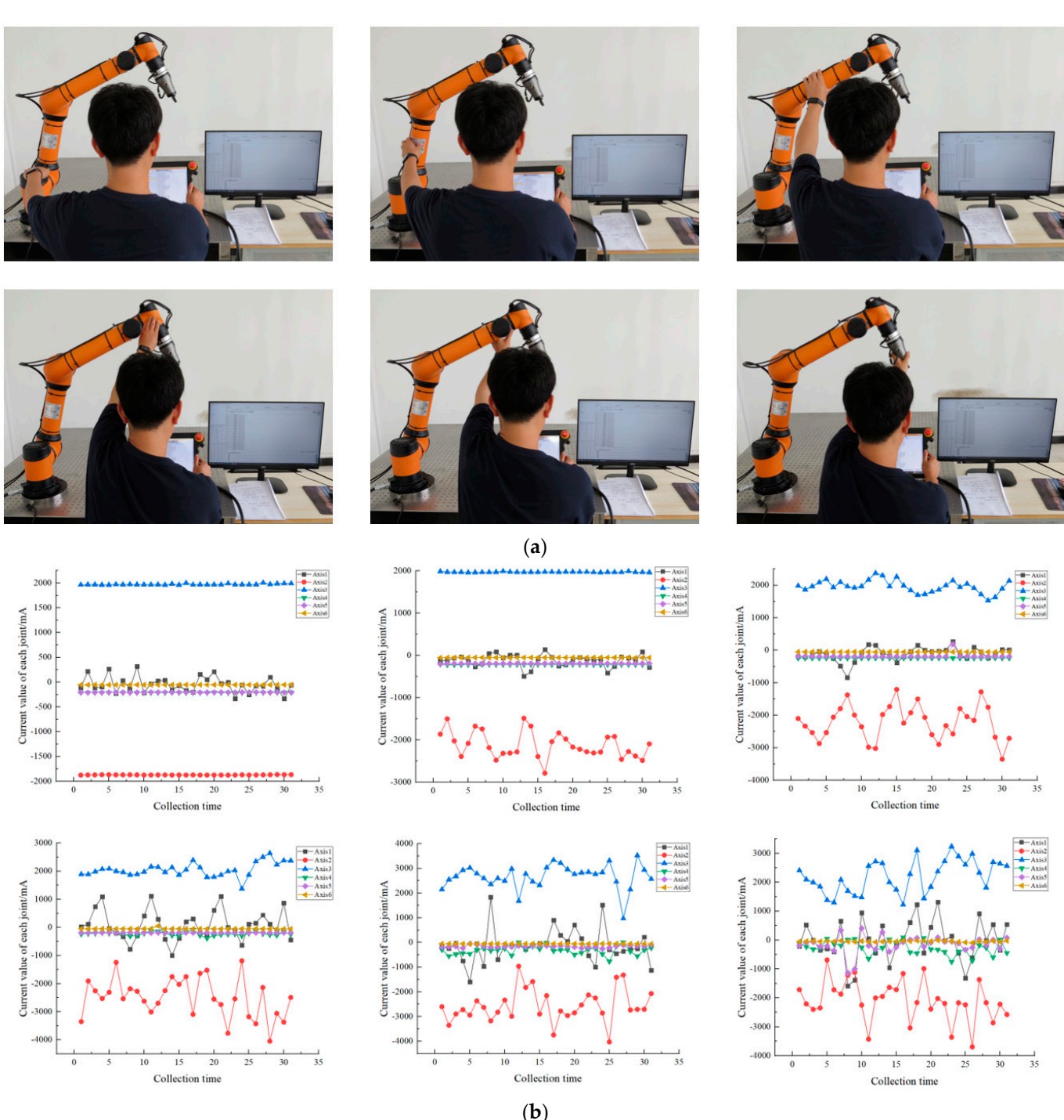

**Figure 9.** Human−robot interaction collision detection experiment. (**a**) Volunteer−simulated collision experiment. (**b**) Joint 1−Joint 6 variations in current data.

The relative error for the actual external contact force is calculated by the formula

$$\Delta \delta F = \frac{\triangle F}{|F_e|} \times 100\% \tag{40}$$

where $\triangle F_n$ is the absolute error of the external contact force and $|F_e|$ is the mode of the applied external force vector. The experimental results obtained by applying different magnitudes of external contact forces at different positions of the robot are shown in Table 2.

**Table 2.** Experimental results of different collision external forces.

| Projects | Location Information/$l'$ | External Forces/N | Computing Power/N | Simulation Power/N | Force Absolute Error/N | Relative Error ofForce/% |
|---|---|---|---|---|---|---|
| 1 | 0.1 | 60 | 32.400 | 35 | 2.600 | 4.333 |
| 2 | 0.1 | 100 | 54.356 | 58.33 | 3.974 | 3.974 |
| 3 | 0.11 | 60 | 32.136 | 32.5 | 0.364 | 0.607 |
| 4 | 0.11 | 100 | 53.350 | 54.167 | 0.817 | 0.817 |
| 5 | 0.12 | 60 | 30.400 | 30 | 0.400 | 0.667 |
| 6 | 0.12 | 100 | 50.324 | 50 | 0.324 | 0.324 |
| 7 | 0.13 | 60 | 26.851 | 27.5 | 0.649 | 1.082 |
| 8 | 0.13 | 100 | 45.118 | 45.833 | 0.715 | 0.715 |
| 9 | 0.14 | 60 | 22.810 | 25 | 2.19 | 3.65 |
| 10 | 0.14 | 100 | 38.400 | 41.667 | 3.267 | 3.267 |

As shown in Figure 10, when a collision occurs, the relative error of the external force on the collision point is 4.333% of the maximum when the position of the collision point is solved by the principle of force law and the principle of virtual displacement, and the accuracy is higher the closer the position of the collision point is to the midpoint of the collision joint of the robot, thus meeting accuracy requirements. Therefore, when the force situation meets the accuracy requirement, the variable $l'$ in the force method principle can be found, and the coordinates of the collision point $P$ can also be found.

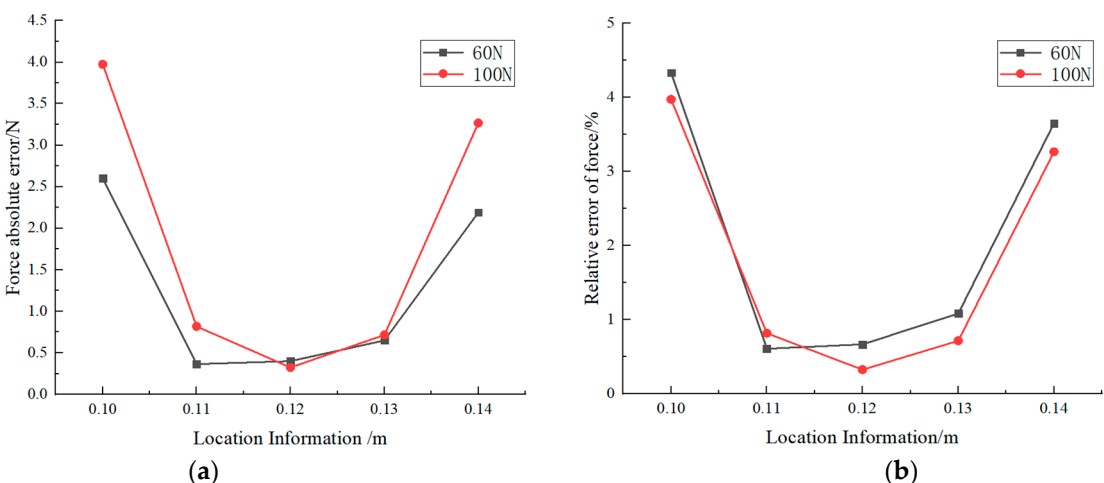

**Figure 10.** Experimental results. (**a**) Force absolute error. (**b**) Force relative error.

## 6. Conclusions

In this paper, we propose a method to establish communication between the robot and the host computer with the aim of clarifying information about joints where accidental collisions occur during robot operation by observing the real-time current changes in each joint axis of the robot. The derivation of the force compensation algorithm is completed by the data information transmitted from the six-dimensional force sensor installed at the robot base. The force law principle and the virtual displacement principle are introduced to solve the specific position information of the specific collision point in the known collision joints. Using the AUBO collaborative robot as the main body of the experiment, experimental verification was performed, experimental data were recorded, and conclusions were drawn through comparative analysis. The algorithm of locking the collision joint is based on

real-time current changes in the robot body and the collision point position information being solved based on the force method principle and the virtual displacement principle, and the error of the force situation is under a reasonable range, which meets the accuracy requirement of the robot collision point identification algorithm.

**Author Contributions:** Conceptualization, Z.W. and B.Z.; methodology, B.Z.; software, B.Z.; validation, Z.W., B.Z. and Y.Y.; formal analysis, B.Z. and Z.W.; investigation, B.Z.; resources, Z.L.; data curation, Z.L.; writing—original draft preparation, B.Z.; writing—review and editing, B.Z. and Z.W.; visualization, Z.L.; supervision, Z.W. and Z.L.; project administration, Z.W. and Z.L.; funding acquisition, Z.W. All authors have read and agreed to the published version of the manuscript.

**Funding:** National Natural Science Foundation of China (No. 51505124), Science and Technology Project of Hebei Education Department (ZD2020151), and Tangshan Science and Technology Innovation Team Training Plan Project (21130208D).

**Institutional Review Board Statement:** Not applicable.

**Informed Consent Statement:** Informed consent was obtained from all subjects involved in the study.

**Data Availability Statement:** The data presented in this study are available on request from the corresponding author.

**Conflicts of Interest:** The authors declare no conflict of interest.

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
