# Peer review of "Research on Identifying Robot Collision Points in Human–Robot Collaboration Based on Force Method Principle Solving"

_actuators, doi:10.3390/act12080320_

Round 1
Reviewer 1 Report
In the paper the principle of identification of robot collision points is proposed. This solution is envisaged in human-robot collaboration based. The force measurement is used to solve the collision.
The proposed solution is suitable for the journal. However some issues may improve the paper quality i.e.:
1. Because the paper is submitted to the journal Actuators more attention should be devoted to discussing the drives used in robots, and in particular the energy, i.e. the current consumed by the drives during operation and collisions.
2. The literature overview quality, in the area of obstacle recognition and control of robots should be improved by adding citations, like:
- A. Gadsden, Y. Song, and S. R. Habibi Novel model based estimators for the purposes of fault detection and diagnosis, IEEE/ASME Trans. Mechatronics, vol. 18, no. 4, pp. 1237–1249, Aug. 2013.
- T. Lindner, A. Milecki, A. Reinforcement Learning-Based Algorithm to Avoid Obstacles by the Anthropomorphic Robotic Arm. Appl. Sci. 2022, 12, 6629. https://doi.org/10.3390/app12136629
- A. Lomakin, J. Deutscher Reliable Algebraic Fault Detection and Identification of Robots, in IEEE Transactions on Automation Science and Engineering, 2022, p. 1 – 17, doi: 10.1109/TASE.2021.3137182
3. The article is very theoretical, so some suggestions for practical solutions regarding the construction of a Six-axis force/torque sensor, its assembly in the robot, etc. should be added to it.
4. How the given in the article theoretical models i.e. equations are used in practical application should be explain. Please specify in chapter 5 which formulas and where were used in the proposed collision detection algorithm.
5. Please add a block diagram of the proposed algorithm, showing its operation. In particular, please specify how the developed system will react to collision detection.
6. The same concerns observing the change of the current of each joint of the robot, and making a judgment whether the robot has an accidental collision. Please explain the sentence in lines 347, 348 “a script program to write the robot system by communicating the robot with the host computer through the network protocol to read the data”
7. The presented in Fig. 8 results should be explained in more detail. The same concerns Fig. 9.
8. Please describe how the algorithm will work in the case of normal robot operation, i.e. when catching, carrying and lowering an element (pick and place operation), or when performing machining operations, e.g. drilling. In such cases, there is no collision, but there are changes in: forces and moments as well as motor currents.
9. The English language of the article should be checked and corrected.
Author Response
Response to Reviewer 1 Comments
Point 1: 1. Because the paper is submitted to the journal Actuators more attention should be devoted to discussing the drives used in robots, and in particular the energy, i.e. the current consumed by the drives during operation and collisions.
Response 1: Combined with the reference changes in question 2, the questions you raised have been noted in the introduction of the article.
Point 2: The literature overview quality, in the area of obstacle recognition and control of robots should be improved by adding citations, like:
- A. Gadsden, Y. Song, and S. R. Habibi Novel model based estimators for the purposes of fault detection and diagnosis, IEEE/ASME Trans. Mechatronics, vol. 18, no. 4, pp. 1237–1249, Aug. 2013.
- T. Lindner, A. Milecki, A. Reinforcement Learning-Based Algorithm to Avoid Obstacles by the Anthropomorphic Robotic Arm. Appl. Sci. 2022, 12, 6629. https://doi.org/10.3390/app12136629
- A. Lomakin, J. Deutscher Reliable Algebraic Fault Detection and Identification of Robots, in IEEE Transactions on Automation Science and Engineering, 2022, p. 1 – 17, doi: 10.1109/TASE.2021.3137182
Response 2: Corresponding changes have been made in the introduction section of the article. At the same time, in response to your suggestions, we have also made changes to improve the quality of citations.
Point 3: The article is very theoretical, so some suggestions for practical solutions regarding the construction of a Six-axis force/torque sensor, its assembly in the robot, etc. should be added to it.
Response 3: The six-axis force / torque sensor mounted at the base of the robot is the KWR200X. It is a multi-dimensional force / torque sensor with a built-in high-precision embedded data acquisition system, which can measure and transmit the orthogonal force and orthogonal moment in three directions in real time. It is made of high-strength alloy steel with strong bending resistance. The assembly method is fixed to the table by bolting, and the robot is fixed to the six-axis force / torque sensor by bolting.
Point 4: How the given in the article theoretical models i.e. equations are used in practical application should be explain. Please specify in chapter 5 which formulas and where were used in the proposed collision detection algorithm.
Response 4: In practical applications, when the software and hardware of the robot system are installed and the program is run. First, the collision joint is locked according to the change in the output joint current. Combined with the force and moment values output by the robot during normal operation and some parameter information of the robot body, it is brought into the equations (23)-(35) to solve the algorithm parameters. In the equations (36)-(38), because the value information of force and moment at the time of collision is known, only the only variable l in the algorithm is an unknown condition, l can be found, and the collision point location can be found. In Chapter 5, the application of the formula is explained in section 5.2, and the formulas (27) through (35) are used to complete the calculations in the table. The location of the simulated collision point in Table 1 and Table 2 of Section 5.2 is the part of the forearm between joint 3 and joint 4 of the robot.
Point 5: Please add a block diagram of the proposed algorithm, showing its operation. In particular, please specify how the developed system will react to collision detection.
Response 5: To run the program, you first need to understand the current values of each joint when the robot works normally. According to the output real-time current value of each joint, it is possible to determine whether an accidental collision has occurred. When an accidental collision occurs, the algorithm parameters are solved according to the parameter information of the robot body and the output value of the force sensor. Then, based on the value of the force sensor at the time of collision, the magnitude of the variable l in the algorithm is solved. At this point, the location of the collision point can be represented. The algorithm block diagram is shown in Figure 8.
Point 6: The same concerns observing the change of the current of each joint of the robot, and making a judgment whether the robot has an accidental collision. Please explain the sentence in lines 347, 348 “a script program to write the robot system by communicating the robot with the host computer through the network protocol to read the data”
Response 6: The meaning of lines 347,348 is: After the robot establishes a data connection with the computer, the script program written begins to run, and the real-time current information at each joint of the robot can be read through the computer.
Point 7: The presented in Fig. 8 results should be explained in more detail. The same concerns Fig. 9.
Response 7: The serial number in the figure has been changed, the original figure 8 has become Figure 9, and the original figure 9 has become Figure 10.
The program runs, as shown in Figure 9(a), the experimental process of volunteers using their hands to simulate the collision experiment and apply external forces to the robot joints 1 to 6, respectively. At the same time, the computer will collect the current at each joint of the robot in six action states and output it to the computer in the form of numerical values. This current information is output in groups every 500ms. Figure 9(b) shows a line chart corresponding to the output of the process of the volunteer hand simulation collision experiment in Figure 9(a). The horizontal axis of this line chart is represented as a set of current information collected every 500ms. The vertical axis of the line chart represents the real-time current situation of each joint of the robot corresponding to each set of current information.
Figure 10 shows the absolute error of the force and the relative error of the force calculated according to equations (39) and (40), and the data are shown in Table 2. Figure 9 shows the corresponding line chart based on the values in Table 2.
Point 8: Please describe how the algorithm will work in the case of normal robot operation, i.e. when catching, carrying and lowering an element (pick and place operation), or when performing machining operations, e.g. drilling. In such cases, there is no collision, but there are changes in: forces and moments as well as motor currents.
Response 8: The significance of the study is that no matter when the robot performs any work such as capture, grasping, placement, drilling, etc., the path and the approximate force at the end of its work are known. When the communication connection is completed and the program is running, the data changes during the normal operation of the robot can be displayed to the staff in real time through the computer. When the robot repeats the above tasks, the staff will get new data information, and by comparing it with the data information during normal operation, the colliding joint can be locked. Then, the proposed algorithm is used to solve the collision point position. For example, when using this system for repeated experiments such as handling, palletizing, assembling, and handling, the first change in data can be used as a known condition. When the same task is repeated, and the forces and moments as well as motor currents data show irregular changes, it can be judged that the robot has accidentally collided with the external environment.
Point 9: The English language of the article should be checked and corrected.
Response 9: Some of the English expressions have been revised.

Reviewer 2 Report
The article deals with collaborative robotics and related work safety due to possible collision situations between humans and robots.
In this paper, the authors proposed a method to identify collision states by detecting the real-time current change in each robot joint.
What kind of 6-axis force sensor was installed, and what are its characteristics?
All calculations and compensations will be handled by a master computer instructing the robot control system to stop immediately during a collision? Can one master computer manage several robots?
Is the proposed procedure generally applicable?
Is any additional intervention required in the robot's control system? Are the standard communication elements of the robot control system sufficient (response speed,...)? Is it possible to use another type of robot? Or even one that is not directly constructed as collaborative?
Figure 8b. The horizontal axis on the graphs are in milliseconds? The units on the horizontal axis are not listed.
Please clearly describe why it is necessary - it is good to know the exact point of the collision when the simplest thing to do in case of detection of a collision (at any point of the robotic arm) is to stop all movement of the arm immediately and put the robot in a state in which it is possible to arbitrarily act on the arm directly change its position manually (the robot accidentally pushes the worker against the wall, so it stops and the worker can move the arm away and not stay pressed).
Typos, eg, line 30 "is-sues", please check elsewhere
The table header is like the last line of the page; the table is on the next page - line 440
References:
The vast majority of Chinese resources and elsewhere in the world are devoted to robotics researchers. It would be good to expand the view of the authors a little.
Author Response
Response to Reviewer 2 Comments
Point 1: 1. What kind of 6-axis force sensor was installed, and what are its characteristics?
Response 1: The KWR200X base six-axis force / torque sensor is a large-range multi-dimensional force sensor with a built-in high-precision embedded data acquisition system that can measure and transmit orthogonal force and orthogonal moment in three directions in real time. It is made of high-strength alloy steel with strong bending resistance. The maximum range of orthogonal forces in three directions is 1000N, and the maximum range of orthogonal moments in three directions is 150N·M.
Point 2: All calculations and compensations will be handled by a master computer instructing the robot control system to stop immediately during a collision? Can one master computer manage several robots?
Response 2: When the robot is stationary or in motion, the experimenter can judge whether the collision has occurred by observing the real-time data information of the force sensor on the computer. When the occurrence of an accidental collision is judged, the robot can press the emergency stop button to abort the execution of the task at the fastest reaction speed. Then, the position information of the collision joint and collision point is locked through the current data in the computer and the collision detection algorithm proposed in the paper.
When the computer and the robot complete the communication settings, it can realize the situation that one computer manages multiple robots. At the same time, it can display the real-time data information of multiple force sensors installed on the robot on a single computer.
Point 3: Is the proposed procedure generally applicable?
Is any additional intervention required in the robot's control system? Are the standard communication elements of the robot control system sufficient (response speed,...)? Is it possible to use another type of robot? Or even one that is not directly constructed as collaborative?
Response 3: After the port and IP address information of different robots and computers in the main program is changed, the assignment, comparison, and data display parts of the joint program are also universally applicable to other systems.
No additional intervention is required.
The robot control system is fine for the output of communication elements such as response speed. If we want to transmit data such as speed and acceleration to a computer in real time, we need to develop it twice.
Combined with the response to the universal applicability of the above procedures, according to different types of robots and force sensors, some of the known equipment parameters in the program and algorithm can be changed for general use for other types of robots. Programs and algorithms are equally applicable depending on the state of the robot and the task it is performing.
Point 4: Figure 8b. The horizontal axis on the graphs are in milliseconds? The units on the horizontal axis are not listed.
Response 4: The order of the article has been adjusted, and the original figure 8 is now Figure 9. In Figure 9b., the horizontal axis represents the output of real-time current values for each joint of the robot every 500 milliseconds.
Point 5: Please clearly describe why it is necessary - it is good to know the exact point of the collision when the simplest thing to do in case of detection of a collision (at any point of the robotic arm) is to stop all movement of the arm immediately and put the robot in a state in which it is possible to arbitrarily act on the arm directly change its position manually (the robot accidentally pushes the worker against the wall, so it stops and the worker can move the arm away and not stay pressed).
Response 5: Because safety is paramount in the task of human-machine collaboration. The contribution of this paper is that when an accidental collision is detected, the accurate location information of the joint and collision point where the collision occurred can be found through the proposed collision detection algorithm. For subsequent equipment repairs and re-routing matters.
Point 6: Typos, eg, line 30 "is-sues", please check elsewhere
The table header is like the last line of the page; the table is on the next page - line 440
Response 6: Modifications have been made to this issue.
Point 7: References:
The vast majority of Chinese resources and elsewhere in the world are devoted to robotics researchers. It would be good to expand the view of the authors a little.
Response 7: The question of references, which has been appropriately changed.

Round 2
Reviewer 1 Report
The article in its current form can be published